# Structural differences between REM and non-REM dream reports assessed by graph analysis

Joshua M. Martin[1¤], Danyal Wainstein Andriano[2], Natalia B. Mota[1], Sergio A. Mota-Rolim[1], John Fontenele Araújo[3], Mark Solms[2], Sidarta Ribeiro[1] *

**1** Brain Institute, Federal University of Rio Grande do Norte, Natal, Brazil, **2** The University of Cape Town, Cape Town, South Africa, **3** Department of Physiology and Behavior, Federal University of Rio Grande do Norte, Natal, Brazil

¤ Current address: Berlin School of Mind and Brain, Humboldt-Universität zu Berlin, Berlin, Germany
* sidartaribeiro@neuro.ufrn.br

**Data Availability Statement:** All relevant data are within the manuscript and its Supporting Information files.

## Abstract

Dream reports collected after rapid eye movement sleep (REM) awakenings are, on average, longer, more vivid, bizarre, emotional and story-like compared to those collected after non-REM. However, a comparison of the word-to-word structural organization of dream reports is lacking, and traditional measures that distinguish REM and non-REM dreaming may be confounded by report length. This problem is amenable to the analysis of dream reports as non-semantic directed word graphs, which provide a structural assessment of oral reports, while controlling for individual differences in verbosity. Against this background, the present study had two main aims: Firstly, to investigate differences in graph structure between REM and non-REM dream reports, and secondly, to evaluate how non-semantic directed word graph analysis compares to the widely used measure of report length in dream analysis. To do this, we analyzed a set of 133 dream reports obtained from 20 participants in controlled laboratory awakenings from REM and N2 sleep. We found that: (1) graphs from REM sleep possess a larger connectedness compared to those from N2; (2) measures of graph structure can predict ratings of dream complexity, where increases in connectedness and decreases in randomness are observed in relation to increasing dream report complexity; and (3) measures of the Largest Connected Component of a graph can improve a model containing report length in predicting sleep stage and dream report complexity. These results indicate that dream reports sampled after REM awakening have on average a larger connectedness compared to those sampled after N2 (i.e. words recur with a longer range), a difference which appears to be related to underlying differences in dream complexity. Altogether, graph analysis represents a promising method for dream research, due to its automated nature and potential to complement report length in dream analysis.

## Introduction

Over the course of a typical night of sleep, the body undergoes characteristic physiological changes, such as variations in brain activity, muscle tone, body shifting and ocular movements.

**Funding:** Authors from Brazil received funding from Coordenação de Aperfeiçoamento de Pessoal de Nível Superior (CAPES; www.capes.gov.br), Conselho Nacional de Desenvolvimento Científico e Tecnológico (CNPq; www.cnpq.br), Financiadora de Estudos e Projetos do Ministério da Ciência e Tecnologia (FINEP; www.finep.gov.br), and Fundação de Apoio à Pesquisa do Estado do Rio Grande do Norte (FAPERN; http://www.fapern.rn. gov.br/). SR was supported by CNPq grants 308775/2015-5 and 408145/2016-1, CAPES-SticAMSud, and Fundação de Amparo à Pesquisa do Estado de São Paulo (FAPESP; www.fapesp.br) grant #2013/07699-0 Center for Neuromathematics. Authors from South Africa received funding from the University of Cape Town (www.uct.ac.za) through fund # 457091 to MS. The funders had no role in study design, data collection and analysis, decision to publish, or preparation of the manuscript.

**Competing interests:** The authors have declared that no competing interests exist.

These changes can be categorized into different sleep stages, each with their own distinctive physiological markers. They include: the state of Rapid-Eye-Movement (REM) sleep and the non-REM sleep stages (sleep onset—N1, light non-REM—N2, and deep non-REM/slow-wave sleep—N3, formerly known as S3 and S4, [1,2].

In addition to the abovementioned physiology, changes in subjective reports of dreaming are also present between the sleep stages. For example, early studies found that awakenings during REM were highly associated with reports of dreaming (~80%), compared to non-REM awakenings (~10%) [3,4]. While this initially led researchers to believe that dreaming was an exclusive property of REM sleep, later studies showed that dream reports could be reliably obtained from non-REM stages [5]. There is now a consensus that dreaming may occur throughout the night during both REM and non-REM sleep; however, disagreement persists over whether dreaming in these distinct phases can be said to be qualitatively different. This point of contention is important, since it has implications for the underlying mechanisms responsible for mental experience during sleep. If the differences are merely quantitative, they suggest that the same underlying mechanism may generate all dreaming experience, only to varying degrees (as claimed by "one-gen theorists", e.g. [6,7]). On the other hand, if qualitative differences are found, it suggests that the processes underlying REM and non-REM dreaming may be driven by distinct mechanisms (as claimed by "two-gen theorists", e.g. [8]). To investigate these possibilities, research over the years has evaluated dream reports collected immediately after laboratory awakenings in REM versus non-REM sleep. Traditionally, this has been done through the use of human judges who rate dreams according to a number of pre-established scales and criteria [9]. Here, we briefly outline some of this previous research.

The first distinction to be noted between REM and non-REM dreaming relates to recall rates, which led to the original controversy about 'REM = dreaming'. An extensive review of 35 studies by Nielsen [10] demonstrated that recall rates are considerably higher in REM (81.9% ± 9.0, mean ± SD), compared to non-REM (43% ± 20.8). However, recall rates for non-REM may vary considerably depending on the sleep stage—dream recall is at its highest during N1 and its lowest during N3.

The second and perhaps most robust difference found between REM and non-REM dreams relates to differing report lengths. The most widely used measure of report length is total recall count (TRC, [6]), which broadly reflects the number of unique words present within a dream report. Studies have consistently found that REM reports are longer than non-REM reports, both when measured in terms of TRC [6,11–14] and when using the raw number of words contained in the report [15–17].

Thirdly, REM and non-REM dream reports tend to differ in their qualitative character. REM reports are typically rated as more intense, bizarre, perceptually vivid, emotional and kinesthetically engaging [8,11,14] than non-REM reports, which are typically more thought-like and conceptual [16,18]. Since REM reports are typically longer than their non-REM counterparts, some authors argue that qualitative measures of REM and non-REM reports can only be meaningfully compared when residual differences in report length are discounted. In this regard, several studies have found that the apparent qualitative differences tend to diminish and even disappear after statistical controls for report length are employed [6,19]. However, even after utilizing such controls, some differences persist [20–22]. Furthermore, the partialling out of report length has been methodologically questioned, since it presupposes that it is the length of a report that causes dream quality and not the other way around [8,23].

A final line of evidence comes from studies comparing REM and non-REM dream reports in terms of their structure, narrative complexity and story-like organization. Nielsen and collaborators [24, 25] found that dream reports collected after REM displayed more of a story-

like organization when compared to reports collected after N2. On the other hand, Cicogna et al. [26] found no difference in the narrative continuity of REM and N2 dream reports obtained from spontaneous morning awakenings; similarly, by using a subsample from this same study [26], Montangero and Cavallero [27] found no differences in a microanalysis of 14 dream reports matched for report length.

While the differences outlined above point to some between-stage differences in dreaming, another important factor to consider is the time of night in which the dream occurs. Throughout a typical night, circadian cortical activation tends to increase, which is associated with characteristic changes in dreaming. Some of these time-dependent changes appear to be common to all sleep phases. For example, both REM and non-REM dream reports become longer [13,20,28], more dreamlike [28, 29], hallucinatory [18] and bizarre [14,30]. However, some of these effects appear to be sleep stage-specific, where, for example, selective increases in emotionality are seen in REM dreaming [14] and a selective decrease in directed thought has been observed in non-REM dreaming [18]. Additionally, the narrative complexity of REM dreams has been found to increase across the night [31,32] although such changes in non-REM dreaming are yet to be investigated.

While previous studies have analyzed the narrative complexity and story-like nature of dream reports, the word-by-word structural organization of REM and non-REM dream reports is yet to be investigated and meaningfully compared. One suitable method for such an evaluation is the *analysis of word graphs*, defined by a given number of nodes (N = 1,2,3. . .) and a set of edges (E = 1,2,3. . .) between them (G = N, E). When the graph represents oral or written discourse, each different word is a node, and the temporal sequence between consecutive words is represented by a directed, unweighted edge. The calculation of mean graph attributes using partially-overlapping sliding windows allows for comparisons across individuals notwithstanding verbosity differences. A non-semantic word-per-node version of this approach has revealed novel behavioral markers of schizophrenia [33,34,35], such as decreased graph connectedness [34] and a more random-like word trajectory [35]. Dream reports appear to be especially revealing of underlying thought disturbances in psychosis [34], and particularly of the negative symptoms of schizophrenia [35]. Graph connectedness has also been shown to predict cognitive functioning and reading ability in typical 6–8 year-olds [36], and to distinguish between elderly patients with Alzheimer's disease, or mild cognitive impairments, and matched controls [37].

Here we investigated the structural organization of REM and N2 dream reports by applying non-semantic word graph analysis to a previously collected sample of dream reports obtained from controlled awakenings in a sleep laboratory. The first aim was to investigate whether REM and non-REM reports are differentially structured in terms of their graph connectedness and distance from a randomly-assembled sequence of words. The second aim was to evaluate how the graph-theoretical method compares to the most widely used measure of report length (i.e. TRC) in dream analysis, and to determine whether or not they can complement one another in this regard. Specifically, we hypothesized that: (1) REM reports will be longer than non-REM reports in terms of report length; (2) REM reports will be structurally different to non-REM reports in terms of graph connectedness and their approximation to random graphs; (3) Graph structure and TRC will change as a factor of the time of night; (4) Graph structure and TRC will be able to discern which sleep stage a dream report was obtained from; and (5) Graph structure and TRC will predict differences in the external ratings of dream complexity (as measured by the Perception Interaction Rating Scale, PIRS).

## Methods

The data were originally collected at the University of Cape Town for the Master's dissertation [38] of author Danyal Wainstein Andriano (DWA). The study used a quasi-experimental repeated measures design whereby participants spent nights in a sleep laboratory to provide dream reports.

### Participants

Twenty-two adults (ages 18–25; mean = 19.71 ± 1.59), all undergraduate Psychology students of the University of Cape Town, were recruited via an online questionnaire to participate in the study. Two participants were excluded due to poor sleep architecture (1) or extreme sleep inertia (1). As a result, dream reports obtained from 20 participants (14 females) were included in the data analysis. Participants were fluent English-speakers (score of 100 or more for the verbal IQ of the Wechsler Abbreviated Scale of Intelligence [39]), reported good sleeping habits (score of 5 or less on the *Pittsburgh Sleep Quality Index* [40]), were moderate to frequent self-reported dreamers (at least once every two weeks [41]), and had no history/presence of illicit substance-use or sleeping/psychiatric disorders.

### Sleep study

The sleep study took place at a hospital sleep laboratory where participants spent 3–4 non-consecutive nights, consisting of one adaptation night, followed by 2–3 experimental nights. During the adaptation night, participants familiarized themselves with the laboratory setting, without controlled awakenings or sleep recordings. On experimental nights, sleep was monitored by polysomnography (PSG) and controlled awakenings were performed in order to obtain dream reports and related questionnaire data. Each experimental night was separated by 2–7 days. This helped minimize any sleep deprivation effects that may have resulted from the experimental awakenings. On the experimental nights, participants arrived at around 19:00 and were prepared for sleep monitoring. DWA switched off the lights at 22:00 and woke the participants at 6:00, totaling approximately 8 hours of sleep recordings per session. Participants were woken for the collection of dream reports 5–6 times over the course of the night, including the morning awakening.

### Awakening protocol

Controlled awakenings were performed in REM, N2 and N3 stages according to the online presence of defining polysomnographic (PSG) characteristics for the respective stages. For REM, the controlled awakenings were conducted 5–10 minutes after detection of muscle atonia (via electromyography; EMG), "saw-tooth" waves in brain activity (via electroencephalography; EEG) and distinct jagged eye-movements (via electrooculography; EOG). For N2 awakenings, the defining criteria included the presence of sleep spindles and K-complexes (via EEG), while N3 consisted of the presence of synchronized, high-amplitude delta waves (via EEG) and diminished muscle tonus (via EMG). In the case of N2 and N3, the length of time spent in a specific sleep stage was not always the same prior to the awakening, since sequences of sleep stability/instability were difficult to predict. At least 40 minutes of uninterrupted sleep was required between awakenings, with at least 15 minutes after a period of REM.

### Dream report collection

When a participant met the defining PSG criteria for the desired stage of sleep, DWA entered the room where the participant was sleeping and called out their name until they verbally

indicated that they were awake. DWA then asked them to recall and report all dream contents that they could remember. The dialogue between participants and DWA was based on the protocol established by Foulkes, Spear & Symonds [42] and Antrobus et al. [30]. Following collection of the verbal dream report, participants were asked to fill out a questionnaire containing a number of Likert scales pertinent to the aims of the original dissertation. Oral dream reports were recorded using a voice recorder and later transcribed and rated by an external judge blind to the conditions of the respective awakenings.

## Word graph analysis

The free software *Speechgraphs* was used to convert transcribed speech into directed non-semantic word graphs (available at: http://neuro.ufrn.br/softwares/speechgraphs, see Fig 1 for an illustration of the transformation). While there are a number of graph measures derived from this analysis, here we chose to evaluate graph connectedness and graph random-likeness, which have been shown to be useful predictors in charting major changes in thought organization, such as those in schizophrenia [34–36]. While both of these reflect aspects of graph structure, they are methodologically distinct and thus have the potential to complement one another in evaluating different aspects of speech structure. Direct evidence for their usefulness as complementary measures can be found in Mota et al. [35], where a linear combination of both connectedness and random-likeness attributes of speech classified negative symptoms and schizophrenia-diagnosis six weeks in advance.

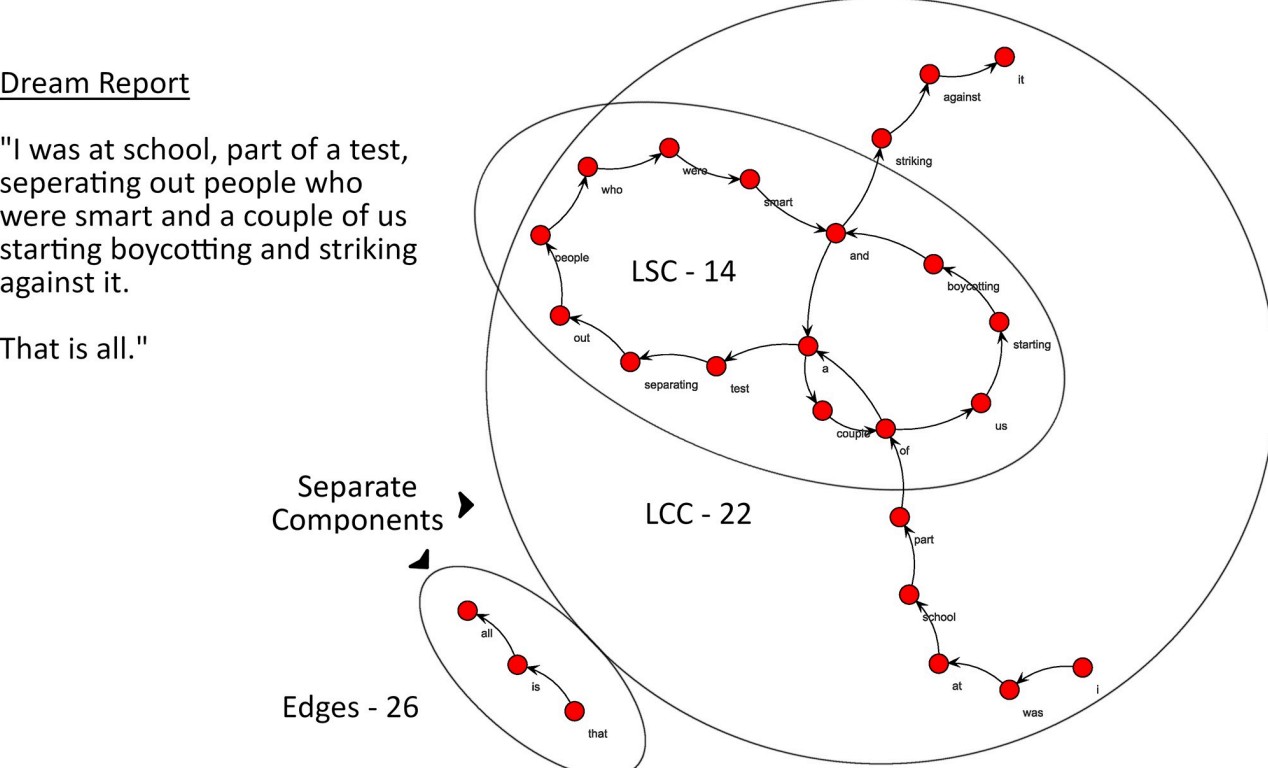

**Fig 1. Word graph analysis applied to dream reports.** Dream report represented as a directed word graph. Nodes indicated in red, edges indicated as black arrows. There are two components in this graph: one with three nodes and the other with 22 nodes. LCC and LSC measures are derived from the larger component.

## Measures of graph connectedness

1. Edges (calculated by the total number of edges present in the graph).

2. Largest Connected Component (LCC; calculated by the number of nodes in the maximal subgraph in which all pairs of nodes are reachable from one another in the undirected subgraph).

3. Largest Strongly Connected Component (LSC; calculated by the number of nodes in the maximal subgraph in which all pairs of nodes are reachable from one another in the directed subgraph, i.e. A leads to B, B leads to A).

## Sliding window to control for report length

Given that connectedness attributes are highly collinear with word count [34], and that REM reports are typically longer than those of non-REM [6], any overall connectedness differences found when using the entire reports in the transformation would be heavily confounded by differences in report length and thus would not be informative. To control for such residual effects, we employed a sliding window method, which controls for word count by dividing the report up according to the window size employed (see Fig 2, for an illustration). A moving window with a fixed length of 30 words and overlap of 29 words was used along each dream report to calculate separate graph measures for each respective window. After reaching the end of the document, the mean value for each measure was calculated across all windows comprised by each report. The window size was based on evidence that 30-word windows are more informative than comparatively smaller sized windows (10 or 20 words; see [34]).

## Comparison with random graphs

To investigate the random-like connectedness of dream reports, we compared each transformed report to 1,000 random graphs, which are assembled using the same number of nodes and edges, but whose word-order is arbitrarily shuffled (Fig 3). Random z-scores for each graph were calculated through subtracting the mean (mrLCC, mrLSC) of the random graph distributions from the original LCC and LSC graph values and dividing the result by their respective standard deviations (sdrLCC, sdrLSC). Graphs that approximate random graphs are those whose z-scores approximate to 0.

## Total Recall Count (TRC)

TRC is an objective measure of report length, which was rated by the researcher, as well as two external judges blind to the awakening conditions. It is measured by the total number of words used to describe any mentation experienced prior to awakening, excluding repetitions, redundancies, "ums" and "ahs", corrections and external commentary on the dream [6]. It is widely used in dream research and known to be one of the best measures to distinguish between REM and non-REM mentation [6]. The measure has been more recently revised under the new name *Word Count Index* [14].

## Perception-Interaction Rating Scale (PIRS)

The PIRS was constructed for the purposes of the original dissertation [38]. The scale was rated by the researcher, as well as two external judges trained to score the dream reports according to an ordinal scale from 0–9, according to the level of *interaction* described between

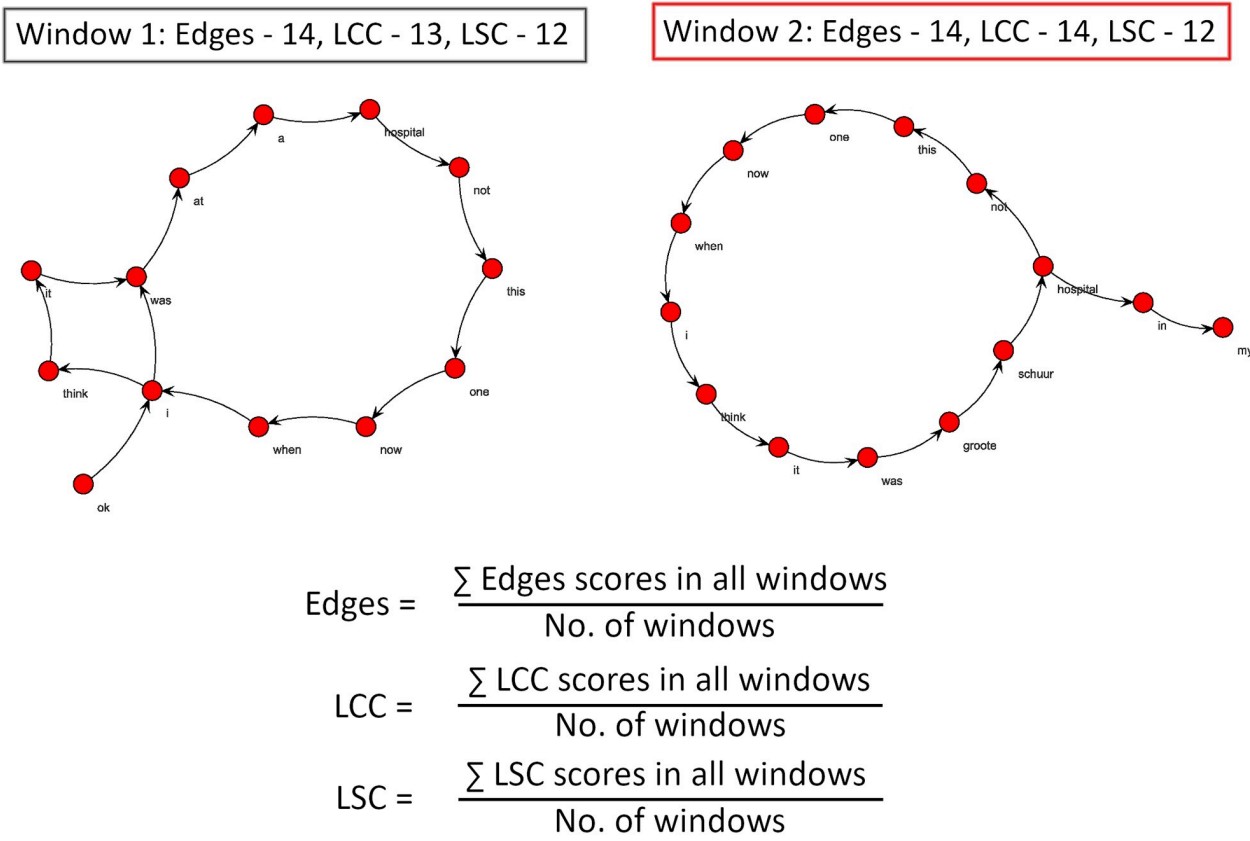

Fig 2. Illustration of the sliding window method. This example uses a window length of 15 words and an overlap of 10 words. While graphs from the first two windows are shown here, the window is applied across the entire dream report, after which an overall average is calculated.

the dream characters and their dream environment. Low scores refer to dreams involving passive unconnected thoughts and imagery, while high scores correspond to dreams involving active engagement with one's environment and include interconnected scenes characteristic of an ongoing narrative (see S1 Text for an overview of the different levels). The scale was developed as a measure of overall dream quality and quantity, and therefore as a proxy of the overall complexity of the dream report.

It is important to distinguish the term "Dream Report Complexity" as it is used here from the term "Network Complexity", which can be used to describe the presence and extent of non-trivial characteristics present in a given graph/network. While both of these terms share the label of "complexity", they describe very different aspects of the dream report that may bear no relation to one another (e.g. a complex dream experience may be verbally reported in such a way that it results in a relatively simple network). To avoid ambiguities, where the term "Dream Report Complexity" is used, we wish to refer to the complexity of the mentation described by the dreamer, as is operationalised by the PIRS. Thus, for the purposes of this article, "dream report complexity" can be considered synonymous with "ratings in PIRS".

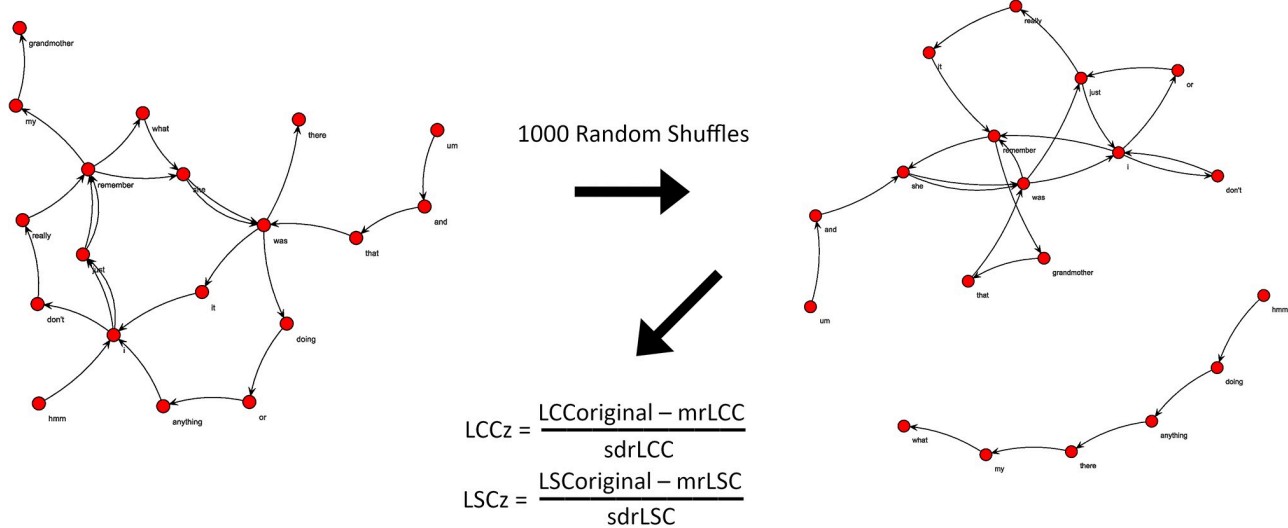

**Fig 3. Illustration of random shuffling.** Word order from the dream report is randomly shuffled 1000 times. The abbreviations "mr" and "sdr" denote the respective mean (mrLCC, mrLSC) and standard deviation (sdrLCC, sdrLSC) scores calculated from this distribution of 1000 shuffled reports. An overall measure of random-like quality is then estimated using the average scores of LCC and LSC based upon this iteration.

## Ethics and informed consent

The study was approved by the Psychology Department's Ethics Committee at Cape Town University prior to data collection. All participants were fully informed about the study, signed consent forms, and were financially compensated for their involvement with R400 (approximately $45 USD at the time of the study) for spending two experimental nights in the sleep laboratory. Participant information was kept strictly confidential. The research and compensation of participants were conducted in accordance with the established guidelines set out by the University of Cape Town's Code for Research and the Helsinki Declaration for human experimentation.

## Data analysis

We performed all analyses in the R environment [43]. Wilcoxon sign-rank tests were used to evaluate differences in REM and non-REM reports, while hierarchical model comparison was used to test the remaining hypotheses. In these cases, generalized linear-models or cumulative link models were compared using the log-likelihood ratio differences of respective models to estimate the significant contribution of individual predictor variables. Models were constructed in a bottom-up manner such that individual predictors are included whose addition significantly improves the fit of the model, following their inclusion. Where applicable, sleep stage as a fixed effect (i.e. REM or N2) is included first as we expect differences in dream reports to exist here based on previous literature. Following this, TRC and variables of graph structure are entered individually to evaluate their respective contribution as predictor variables. Where significant predictors are found, composite models are then considered to

evaluate whether measures may complement one another in predicting the outcome variable. To control for the independence of observations, participant medians were used for Wilcoxon sign-rank tests, and mixed effects models were used to model random effects across participants and experimental nights. To evaluate potential confounds, two confirmatory analyses were run to evaluate the influence of the presence of common words as well as the overall number of paragraphs present in the report. A table reporting the correlations between predictors and other variables of interest can be found in the supplementary material (see S1 Table).

## Results

### Dream recall and report complexity

A total of 198 controlled awakenings were performed during REM and N2 sleep, resulting in the collection of 146 dream reports from 20 participants (see Fig 4 for an overview). Dream recall was more prevalent following REM awakenings (90.74% vs. 72.39), while following N2 awakenings participants were more likely to report having not dreamt (19.40% vs. 7.41%) or to have had a white dream (15.67% vs. 1.85%)—an experience where subjects feel as if they were dreaming but are unable to recall any content. For the final sample in our analysis, 13 dream reports (REM = 3; N2 = 10) were excluded, as they did not meet the minimum word count of 30 words. This resulted in a final sample of 133 reports (N2 = 87; REM = 46). The elevated proportion of N2 reports in our sample reflects the greater number of awakenings that were performed in N2, since non-REM dreaming was the main interest of the original protocol [38]. Of the 133 dream reports utilized in the final sample, those obtained from N2 awakenings more often described isolated visual imagery (42.53% vs. 15.21%) or conceptual, non-visual

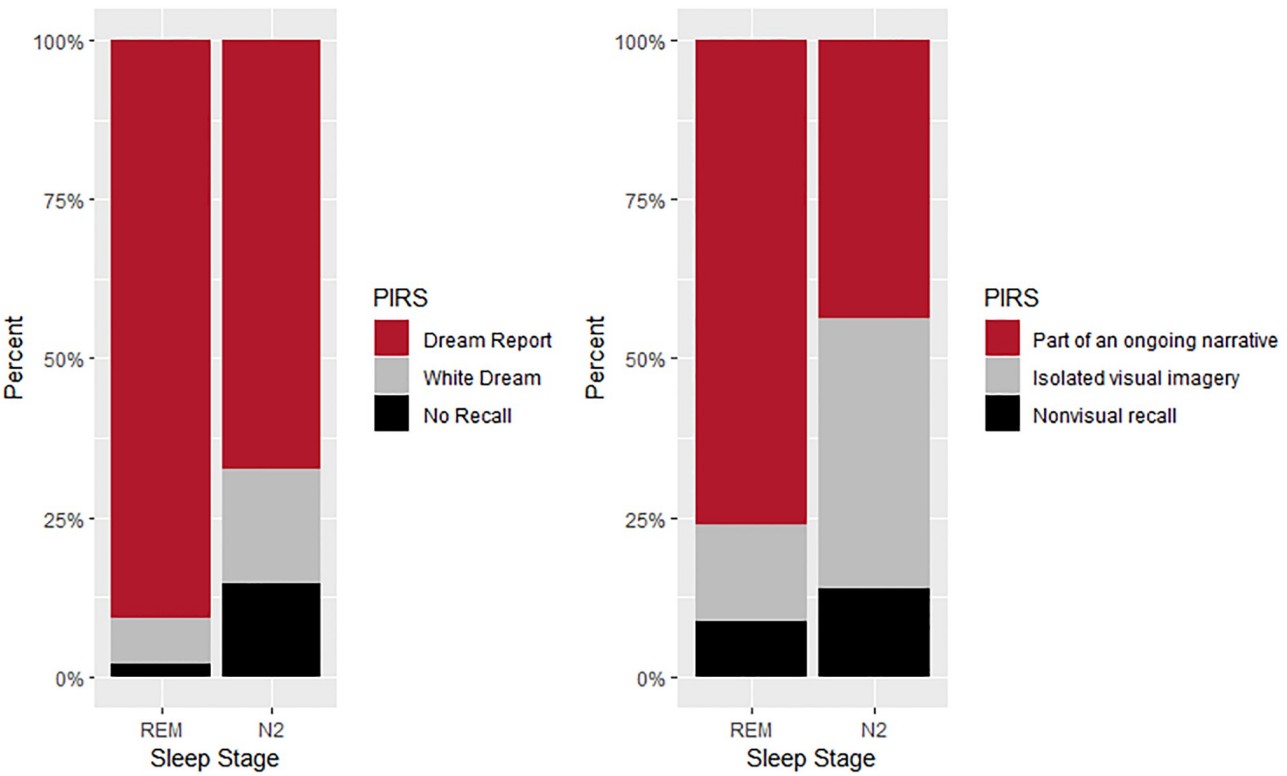

**Fig 4. Stacked bar plot showing prevalence of dream reports and type of mentation recalled.**

**Table 1. Results from Wilcoxon sign-rank tests (n = 40).**

|  | REM | N2 | Z-score | effect size (r) | p-value |
|---|---|---|---|---|---|
| TRC | 51.50 ± 41.00 | 34.75 ± 13.31 | -3.29 | .533 | .001 |
| Edges | 28.65 ± 0.63 | 28.37 ± 0.60 | -2.13 | .346 | .033 |
| LCC | 23.70 ± 1.08 | 22.41 ± 1.15 | -3.19 | .517 | .001 |
| LSC | 16.67 ± 2.61 | 16.10 ± 2.44 | -1.97 | .320 | .048 |
| LCCz | 1.47 ± 1.00 | 1.36 ± 0.37 | -0.68 | .110 | .498 |
| LSCz | 3.76 ± 0.86 | 3.66 ± 0.99 | -0.38 | .062 | .701 |

Values that reach statistical significance ($\alpha < .05$) are shown in bold.

experiences (13.79% vs. 8.70%)., while those obtained from REM awakenings described more elaborate dream sequences indicative of an ongoing narrative (75.09% vs. 43.68%).

## REM vs. N2 differences in graph structure and TRC

We first aimed to investigate differences between REM and non-REM reports. Wilcoxon sign-rank tests were used to compare the participant medians obtained in REM and N2 (see Table 1). We found that REM reports had significantly higher Edges, LCC, LSC and TRC scores compared to N2 reports, a difference with a moderate to large effect size. No significant differences in random-likeness were observed between REM and N2 (i.e. LCCz and LSCz).

## Testing for time of night effect

We next investigated whether TRC and graph measures (Edges, LCC, LSC, LCCz, LSCz) could predict the time of night in which dream reports were obtained. This corresponds to checking for a time of night effect. We first entered sleep stage as a variable for model comparison, since we were interested in whether changes across the night are observed independent of any residual differences that exist between the sleep stages. As a result, variables of interest (Edges, LCC, LSC, TRC, LCCz, LSCz) were entered individually to a model containing sleep stage, to investigate whether their addition improved the overall fit of the model. From the resultant models, none of the variables were found to significantly improve the overall fit (see Table 2). Thus, no time of night effect was found in the present data for any of the respective predictor variables.

**Table 2. Results from generalised linear mixed models in predicting time of night.**

| Individual Predictors | Pseudo $R^2$ | Pseudo $R^2$ Change | p |
|---|---|---|---|
| Sleep Stage | .011 | .011 | .229 |
| Sleep Stage + **TRC** | .022 | .011 | .228 |
| Sleep Stage + **Edges** | .016 | .006 | .386 |
| Sleep Stage + **LCC** | .021 | .010 | .240 |
| Sleep Stage + **LSC** | .011 | <.001 | .960 |
| Sleep Stage + **LCCz** | .013 | .002 | .615 |
| Sleep Stage + **LSCz** | .030 | .019 | .107 |

*Pseudo $R^2$ change values are calculated in comparison to a model containing *sleep stage*, while Pseudo $R^2$ are calculated in relation to the null model. Time of night is measured according to minutes elapsed since lights off (i.e. 22:00 PM). Where applicable, Pseudo $R^2$ change and p-values reflect the contribution of the predictor highlighted in bold.

**Table 3. Results from generalised linear mixed models in predicting sleep stage.**

| Individual Predictors | Pseudo $R^2$ | Pseudo $R^2$ Change | p |
|---|---|---|---|
| TRC | .095 | .095 | .002 |
| Edges | .011 | .011 | .307 |
| LCC | .069 | .069 | .009 |
| LSC | .002 | .002 | .676 |
| LCCz | <.001 | <.001 | .804 |
| LSCz | .011 | .011 | .313 |
| **Composite Models** | Pseudo $R^2$ | Pseudo $R^2$ Change | p |
| TRC + **LCC** | .138 | .048 | .033 |
| LCC + **TRC** | .138 | .074 | .007 |

Values that reach statistical significance ($\alpha < .05$) are shown in red. Significance testing and Pseudo $R^2$ are calculated in comparison to the Null Model for the first set of individual measures, and calculated in comparison to a model containing either TRC or LCC in the composite analyses. Where applicable, Pseudo $R^2$ Change and p-values reflect the contribution of the predictor highlighted in bold.

## Distinguishing sleep stage based on graph structure and TRC

**Testing individual measures.** To test how graph structure compares to TRC as a means to discern sleep stage, we constructed generalised linear models with a binomial (REM/N2) outcome, to examine whether aspects of graph structure could significantly distinguish between reports obtained from REM and N2 sleep and how they may relate to the widely used measure of TRC in this regard. The analysis found that the addition of LCC and TRC significantly improved a null model in predicting differences in REM and N2 (Table 3). The differences after adding Edges, LSC, LCCz, and LSCz were not found to be significant. Thus, mirroring the differences found in our Wilcoxon-sign rank tests, we found that TRC and LCC were the best performing variables in detecting differences amongst REM and N2 reports; however, unlike before, Edges and LSC were not found to be significant predictors in this regard.

**Testing for complementary measures.** We next investigated whether LCC and TRC could act as complementary measures to one another in the discernment of sleep stage. In this regard, we tested whether the addition of LCC to a model containing TRC would significantly improve the fit of the model in predicting differences in sleep stage. The model containing both TRC and LCC was found to be significantly better at predicting sleep stage than TRC alone (Table 3). We performed the same analysis, this time seeing whether TRC could add significantly to a model containing LCC. Once again, the difference between the models was significant, indicating that TRC and LCC are complementary measures in discerning sleep stage.

## Testing the relationship to dream report complexity

**Testing individual variables.** We next evaluated whether TRC and measures of graph structure are related to external ratings of dream complexity (i.e. PIRS). The null model adopted for comparison contained the fixed effect of sleep stage, since we are interested in whether the explanatory variables can significantly improve the fit of the model over and above differences in complexity between the sleep phases.

Table 4 shows that the addition of Edges, LCC, TRC and LCCz to a model containing sleep stage significantly improved the fit of the model in predicting PIRS scores for these variables, while LSC showed a significant trend in the same direction. LSCz was not found to be

**Table 4. Results from cumulative link models in predicting PIRS ratings.**

| Individual Predictors | Pseudo $R^2$ | Pseudo $R^2$ Change | p |
|---|---|---|---|
| Sleep Stage | .138 | .138 | <.001 |
| Sleep Stage + **TRC** | .588 | .522 | <.001 |
| Sleep Stage + **Edges** | .194 | .065 | .003 |
| Sleep Stage + **LCC** | .228 | .105 | <.001 |
| Sleep Stage + **LSC** | .179 | .048 | .012 |
| Sleep Stage + **LCCz** | .138 | <.001 | .858 |
| Sleep Stage + **LSCz** | .171 | .038 | .025 |
| Composite Models | Pseudo $R^2$ | Pseudo $R^2$ Change | p |
| Sleep Stage + TRC + **Edges** | .590 | .005 | .430 |
| Sleep Stage + TRC + **LCC** | .620 | .079 | .001 |
| Sleep Stage + TRC + **LSC** | .591 | .007 | .336 |
| Sleep Stage + TRC + **LSCz** | .620 | .078 | .002 |
| Sleep Stage + TRC + LSCz + **LCC** | .629 | .023 | .090 |
| Sleep Stage + TRC + LCC + **LSCz** | .629 | .023 | .094 |

Values that reach statistical significance (α < .05) are highlighted in red. Values of Pseudo $R^2$ Change are calculated in comparison to the sleep stage model for individual measures and in comparison to the model containing TRC and sleep stage for the composite ones. Where applicable, Pseudo $R^2$ Change and p-values reflect the contribution of the predictor highlighted in bold.

statistically significant. In terms of the direction of this relationship, the results indicated that report length and graph connectedness increases while graph random-likeness decreases in relation to increased ratings of dream report complexity. The effect sizes of graph structure measures, as estimated by a change in Nagelkerke's pseudo-$R^2$, were found to be of a small to medium size; the effect size for the addition of TRC was large. In order to test whether the slope of effect in predicting dream report complexity was different in REM or N2, we tested for the presence of an interaction effect between sleep stage and the fixed effects in the respective models (TRC, Edges, LCC, LSC, LCCz, LSCz). The addition of the interaction effect significantly improved the fit for only Edges (Pseudo $R^2$ Change = .036, p = .029), but not for any of the other measures (*TRC*: Pseudo $R^2$ Change = .016, p = .161; *LCC*: Pseudo $R^2$ Change = <.001, p = .803; *LSC*: Pseudo $R^2$ Change = .005, p = .437; LCCz: Pseudo $R^2$ Change = .004, p = .463; LSCz: Pseudo $R^2$ Change = .015, p = .162). We may therefore assume that, except in the case of Edges, the trends for REM and N2 groups were not significantly different from one another in their prediction of dream report complexity.

**Testing complementary measures.** Given the significant relationships found, we next sought to investigate whether attributes of graph structure that were previously found to be significant could act as complementary measures to TRC in explaining dream complexity. To do so, we compared the log-likelihood ratios of a model containing TRC and the individual connectedness measures to a model only containing TRC. We found that the addition of LCC and LSCz significantly improved the fit of the model; no such effect was found for Edges or LSC. As a result, this suggests LCC and LSCz can act as a complementary measure to TRC in explaining differences in dream report complexity. We then took a final step to evaluate whether LCC and LSCz entered together could further improve the fit of these composite models. Neither model comparison was found to significantly improve the overall fit, although both showed a trend towards significance (0.05 < p < 0.10).

## Dependence on PIRS in predicting sleep stage

Given that our results indicate that LCC and TRC can predict differences in sleep stage (REM vs. N2), and that both are related to measures of dream report complexity, we added a supplementary hypothesis that sought to investigate whether the ability of LCC and TRC to discern between REM and N2 reports is independent of differences in PIRS ratings. By comparing the log-likelihood ratios of the respective models, we found that the addition of either LCC (Pseudo $R^2$ Change = .018, p = .197), TRC (Pseudo $R^2$ Change = < .001, p = .928) or both LCC and TRC (Pseudo $R^2$ Change = .019, p = .432) did not significantly improve the fit of a model containing the predictor of PIRS in sleep stage discernment. This suggests that once differences in dream report complexity are partialled out, both TRC and LCC are unable to statistically distinguish between REM and N2 dream reports.

## Follow-up analyses: Controlling for common words and number of paragraphs

Following our main analysis we performed two follow-up analyses to evaluate the effects of two potential confounds to our results. Firstly, given that graph loops are often intersected by common pronouns, prepositions and conjunctions, we sought to investigate whether the above results can be explained merely by the increased occurrence of these classes of words in certain dream reports. To do this, we applied a standard list of English NLTK stop-words (accessed via https://gist.github.com/sebleier/554280) to the dream reports and re-evaluated the analyses where graph attributes were found to be significant predictors (see S1 Appendix). We were able to reproduce our findings above with comparable results: despite a reduced sample size (n = 113), all findings were either still found to be significant, or still demonstrated a trend towards significance in the same direction. Effect sizes were comparable to before, and in some cases were found to be even stronger (e.g. in predicting dream report complexity).

Secondly, given that LCC and Edges scores are affected by distinct graph components, we next sought to rule out the possibility that our findings may merely be reflected by differences in the overall number of paragraphs, deriving from different turn-taking between the participant and researcher. Indeed, when comparing the average number of paragraphs we found that, on average, N2 reports had on average more paragraphs (median = 3.5, standard deviation = 1.75) than REM ones (median = 2.5, standard deviation = 2.09). This raises the possibility that differences in graph structure may merely reflect an increased occurrence in the number of paragraphs in N2 dream reports. To control for this confounding influence, we performed a supplementary analysis partialling out the number of paragraphs before evaluating the ability of graph measures to predict differences in sleep stage and dream report complexity (see S2 Appendix). Once again we were able to reproduce our main results: the addition of graph structure predictors was still found to improve the overall fit of the respective models. There were two exceptions: LCC failed to complement TRC in predicting sleep stage, and LSCz failed to predict differences in dream report complexity. Nonetheless, both of these cases demonstrated a significant trend in the same direction (LCC, p = .062; LSCz, p = .061). Effect sizes were slightly reduced, which is not surprising given that any shared explanatory variance between graph structure measures and the number of paragraphs would have been partialled out by the control analysis. Overall, given that our core findings were replicated, we interpret this to rule out common words or differences in paragraphs as potential confounds to the present results.

## Discussion

Here we investigated differences in the structural organization of REM and non-REM dream reports, and how structural non-semantic graph measures may compare to report length (i.e. TRC) in dream report analysis. This is the first study to demonstrate that when represented as graphs, *REM dream reports possess a larger structural connectedness compared to N2 reports*, a result that cannot be explained by differences in report length. It also indicates that graph structure, both in terms of connectedness and its random-likeness, is informative of dream report complexity, where *more complex dreams are associated with larger connectedness and less random-like graph structures*. Finally, the results demonstrate that aspects of graph connectedness (specifically LCC and LSCz) can act as *a complementary measure to TRC in predicting differences in REM and non-REM dream reports and overall ratings of dream complexity*. Collectively, our results complement the existing literature reporting qualitative differences in REM and non-REM dream reports, and point to non-semantic graph analysis as a promising automated measure for future use in dream research.

### REM reports are longer and have larger connectedness compared to N2

The results of the present study are consistent with findings in previous studies pointing to overall differences in REM and non-REM dream reports. Firstly, we found that dream recall is higher in REM than N2 awakenings [10]. Secondly, we found that qualitatively, REM dreams were more part of an ongoing narrative while non-REM dreams involved non-visual, conceptual recall. This is consistent with previous studies showing that REM dreams are more hallucinatory [18] and story-like [25] while non-REM dreams are often thought-like [18] and conceptual [16]. Finally, in our sample, REM reports were typically longer than N2 ones (i.e. higher TRC), supporting previous studies showing that one of the most robust differences between these two groups relates to report length [6].

Through using a sliding window method, to control for differences in report length, we aimed to investigate whether intrinsic structural differences are found between these reports from REM and N2. The results showed that REM reports had larger connectedness compared to N2 in terms of LCC, Edges and LSC with moderate to large effect sizes. On the other hand, when comparing dream reports to those that were randomly shuffled 1,000 times, we did not find any differences in REM and non-REM reports in their random-likeness. This suggests that, on average, words contained in REM reports tend to recur with a longer range compared to those in N2 reports, forming longer loops and far-reaching connections, resulting in larger connectedness. However, they suggest that these structural differences are not accompanied by differences in the way that they approximate to random speech, such as is found in people suffering from schizophrenia [35]. In terms of a time of night effect, we were not able to replicate findings from previous studies [14,30], which demonstrated changes in qualitative and quantitative aspects of dream reports across the night. In our study, both graph measures and TRC did not change as a factor of the time of night. Given that TRC has been found to change significantly across the night [11,13], it is unclear whether the findings for graph structure here reflect a genuine null effect or a particular characteristic of our sample. Given that controlled awakenings were also conducted during N3 in our sample, we speculate that sleep deprivation from numerous awakenings may have displaced sleep architecture, resulting in changes to the characteristic sleep cycle needed for a time of night effect to occur.

These results collectively suggest that dream reports are less frequent in N2, and when they are present, they are typically shorter, more thought-like and have smaller connectedness compared to their REM report counterparts. Given that many differences in REM and non-REM reports are highly diminished or even disappear after controlling for length [6], these findings

also have value in supplementing the small group of studies that have found differences between these sleep stages over and above residual differences in report length [20–22]. Further research may investigate the time of night effect, in order to clarify whether graph connectedness increases across the night in a similar fashion to other dreaming variables reported in previous studies [14,30].

## Graph connectedness in relation to dream reports across the sleep cycle

Previous studies have found that graph measures from dream reports can be particularly informative of the thought disturbances that underlie psychosis [33,35]. Such findings naturally prompt comparisons to the long-held phenomenological comparisons [44,45] of dreaming as a model for psychosis [34,46]. One of the hallmark differences between REM and non-REM dreaming is the more bizarre, hallucinatory nature of the former [18]. By extension, one may speculate that graphs obtained from REM reports would be more closely related to those of people with schizophrenia (i.e. would be less connected). However, such an interpretation is contradicted by the present findings, where REM graphs had on average *larger* connectedness compared to N2 graphs, and not the other way round. If we were to apply this framework to our sample, it would suggest that N2 dream reports mimic the reports of those with psychosis more than REM reports do, which seems improbable according to its phenomenology. Thus, while the phenomenological aspects of dreaming may approximate the experiences of people with psychosis, the differences in the connectedness of dream reports across the sleep cycle in healthy young adults do not reflect this.

We believe a more suitable approach to the present data would be to interpret the observed differences in graph connectedness in terms of variations in the cognitive ability of participants to retrieve and organize their dream experiences. This is in accordance with findings that graph connectedness tends to increase in healthy cognitive development in children [36] and declines in age-related dementias [37] and some psychopathologies [33–35] where cognitive impairment is commonly observed.

For the present study, we postulate that the observed changes of graph connectedness in dream reports across the sleep cycle may be conceivably affected by two main factors. The first factor is related to sleep inertia and the immediate effects upon cognition of the sleep/wake transition, whereby memory and attention processes may be impaired. Since sleep inertia is more marked in N2 compared to REM [47], one can imagine that this may exert a more negative impact on the ability to mentally organise one's thoughts in N2, leading to the decrease in report connectedness as compared to REM.

The second factor is related to the nature of the dream experience itself. Since the quality of dreaming may vary considerably, both within and between sleep states, it is possible that the ability to organize experience into a verbal report may be influenced by the underlying complexity of the dream experience to be described. In this sense, dream experiences that are coherent, immersive and story-like may be more easily organized into a report with larger connectedness, while dream experiences that are fragmented and isolated are relatively more difficult to organize mentally and thus are structurally less connected. While complex dream narratives may occur in N2, REM physiology may provide more favourable conditions for such dreams to occur, given the diffuse cortical activity and increased activation of the motor cortex [48] coupled with muscle atonia, allowing for an immersive, interactive narrative to develop uninterrupted.

To estimate the relative contribution of these two processes, three findings are of potential interest. Firstly, once we partialled out differences in PIRS ratings, we found that LCC could no longer distinguish between REM and N2 dream reports. Secondly, by using a model

containing sleep stage as a statistical comparison, we showed that graph connectedness could significantly predict PIRS over and above any differences in sleep stage (i.e. when graph differences related to the sleep stage are partialled out). Finally, with the exception of Edges, no significant interaction effect was found between the graph attributes and sleep stage as a variable, indicating that the modeled relationship between TRC and graph connectedness with PIRS was largely comparable for both REM and N2 dream reports.

On the surface these results appear to argue against the role of sleep inertia, since graph connectedness is more closely related to differences in ratings of dream complexity than it is to differences between the REM and N2 sleep stages. However, the PIRS ratings themselves may be confounded by sleep inertia, since they too are based upon verbal reports collected after awakening. Given this possibility, the role of sleep inertia cannot be ruled out as an explanation for the present findings. To tease apart the relative contribution of these two processes, future research should investigate the relationship between the narrative/story-like complexity of dreams and their graph connectedness in different samples. Since the narrative complexity of dream reports persists even after a period of time has elapsed [31], one may uncouple the effects of sleep inertia from dream complexity through analysing and comparing the story-likeness and structural connectedness of reports obtained immediately after awakenings to another set of reports that describe the same dream experiences during the night, after a delay, where any residual cognitive effects of the sleep/wake transition should be greatly diminished. Clearly, since the two explanations are not mutually exclusive, graph connectedness is likely to be affected by a combination of these factors, as well as other factors not considered here.

## Graph analysis as a method for dream research with clinical potential

By utilizing hierarchical model construction in discerning sleep stage (REM vs. N2) and levels of dream complexity (as measured by the PIRS), we were able to probe how graph connectedness compared to TRC in modeling these variables of interest and whether it could act as a complementary measure in this regard. We found LCC could predict differences in sleep stage and could significantly improve a model containing TRC in this prediction, albeit with a small effect size. We also found that individually LCC and LSCz could significantly improve a model containing TRC in predicting ratings on the PIRS. Given that TRC is one the most widely used measures to distinguish REM and non-REM reports, this finding is of particular important since it suggests that graph-based analyses of report structure may act as a complementary measure to TRC in discerning the sleep stage of a report and measuring underlying aspects of dream complexity. While Edges and LSC did not significantly discern REM and non-REM dreams or significantly improve models containing TRC, they still showed promise in predicting differences in dream report complexity.

As a whole, these findings point to non-semantic graph analysis as a potentially valuable tool for dream report analysis. The automated nature of this analysis means that it is fast, low-cost and avoids the biases and problems of reliability inherent in methods that involve human rating systems [9]. It offers a number of methodological advantages, as it may be applied to large corpora of dream reports that may otherwise be too time-consuming and/or expensive to apply traditional, human-based rating systems. The advent of the *Dream Bank* [49], which now holds more than 20,000 dream reports represents an example where computational methods such as non-semantic graph analysis may hold particular value.

The present study also extends and corroborates previous findings on the non-semantic graph structure of dream reports in healthy controls, which differs substantially from the structures observed in dream reports from patients with schizophrenia or Alzheimer's disease [33–37]. Exploration of the clinical implications of the method must include the assessment of

patients with various non-REM or REM sleep disorders, as well as a fine-grained comparison of the effects of psychiatric medications on the structure of dream reports.

## Limitations and future perspectives

In light of the present findings, a number of limitations need to be considered. Firstly, it is unclear how sleep inertia may have affected the graph connectedness results. While we have shown statistically that such an influence is unlikely to fully explain differences in graph connectedness, it cannot be ruled out. Secondly, our participant median TRC estimates in REM (51.5) and N2 (34.75) are closer to one another compared to those cited in previous studies (e.g. [11] REM—40, N2–13; [12] REM—148, N2–21). Thus, it is possible that TRC's potential as a measure to predict differences in sleep stage may be diminished here, due to inherent characteristics of the sample. Finally, while we have reported differences in REM and non-REM reports, the scope of our non-REM findings is restricted to N2 reports. Future studies incorporating N1 and N3 reports, as well as waking mentation reports, should enhance our understanding of these changes across the sleep/wake cycle in relation to underlying mentation.

## Conclusions

We have shown that the word-to-word structural organization of dream reports is informative about the sleep stage in which it was obtained and the overall complexity of the dream report, even when differences in report length are partialled out. Our results are consistent with previous findings showing that dreaming in N2 as compared to REM is less frequently recalled and, when present, is shorter, less intense and more thought-like and conceptual. Our results also supplement previous research by showing that N2 reports display smaller connectedness (i.e. words recur over a shorter range) compared to their REM report counterparts. Although a time of night effect has been found in previous literature, we were not able to replicate the finding here, possibly due to the displacement of deep sleep due to multiple experimental awakenings in N3. While the effects of sleep inertia cannot be ruled out, the observed differences in graph structure appear to reflect underlying differences in the dream complexity, where coherent, story-like dream experiences (more commonly found in REM), are more likely to be organized with larger connectedness and less random-like report structure. These findings represent a significant step towards characterizing the evolution of the structure of mentation across the various phases of the sleep cycle. They also point to non-semantic graph analysis as a promising automated measure for sleep research due to its sensitivity to dream complexity and its ability to complement report length in the analysis of REM and non-REM dream reports. Further research can replicate and extend these findings through clarifying the effects of sleep inertia on graph connectedness and evaluating the evolution of graph structure according to the time of night effect. Such investigations can enhance our knowledge of dreaming and its various manifestations throughout the night, while providing additional evidence for the application of automated graph-based methods in dream research.

## Supporting information

**S1 Text. Overview of levels of Perceptual Interaction Rating Scale (PIRS).**
(DOCX)

**S1 Table. Correlation matrix showing relationship between variables of interest.**
(DOCX)

**S1 Appendix. Showing results for follow-up analysis controlling for the occurrence of common words (via NLTK list).**
(DOCX)

**S2 Appendix. Showing results from follow-up analysis partialling out number of paragraphs.**
(DOCX)

## Acknowledgments

We would like to thank Mariza van Wyk and Michelle Henry for their help in evaluating the dream reports as external judges and Gal Adar and Tony Friend for the motivational support in the writing of the manuscript.

## Author Contributions

**Conceptualization:** Joshua M. Martin, Danyal Wainstein Andriano, Mark Solms, Sidarta Ribeiro.

**Data curation:** Joshua M. Martin, Danyal Wainstein Andriano, Mark Solms, Sidarta Ribeiro.

**Formal analysis:** Joshua M. Martin, Danyal Wainstein Andriano, Natalia B. Mota.

**Funding acquisition:** Mark Solms, Sidarta Ribeiro.

**Investigation:** Joshua M. Martin, Danyal Wainstein Andriano, Sergio A. Mota-Rolim, Mark Solms, Sidarta Ribeiro.

**Methodology:** Joshua M. Martin, Danyal Wainstein Andriano, Natalia B. Mota, John Fontenele Araújo, Mark Solms, Sidarta Ribeiro.

**Project administration:** Sidarta Ribeiro.

**Resources:** Joshua M. Martin, Mark Solms, Sidarta Ribeiro.

**Software:** Joshua M. Martin.

**Supervision:** Mark Solms, Sidarta Ribeiro.

**Validation:** Joshua M. Martin, Danyal Wainstein Andriano, Natalia B. Mota.

**Visualization:** Joshua M. Martin, Natalia B. Mota, Sidarta Ribeiro.

**Writing – original draft:** Joshua M. Martin, Sidarta Ribeiro.

**Writing – review & editing:** Joshua M. Martin, Danyal Wainstein Andriano, Sergio A. Mota-Rolim, John Fontenele Araújo, Mark Solms, Sidarta Ribeiro.

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
