## [Decision Letter · Decision Letter 0]

18 Mar 2020

PONE-D-20-01935

Structural differences between REM and non-REM dream reports assessed by graph analysis

PLOS ONE

Dear Dr. Ribeiro,

Thank you for submitting your manuscript to PLOS ONE. After careful consideration, we feel that it has merit but does not fully meet PLOS ONE’s publication criteria as it currently stands. Therefore, we invite you to submit a revised version of the manuscript that addresses the points raised during the review process.

ACADEMIC EDITOR: 

The reviewer raised issues about the description and also the choice of the methodology proposed by the authors. You have to describe in a more scientific way the pipeline of the proposed methodology and also you have to justify every choice that you made in the proposed graph-based analysis.

I personally agree with the comments and I suggest to revise it according to his/her guidelines.

We would appreciate receiving your revised manuscript by May 02 2020 11:59PM. To enhance the reproducibility of your results, we recommend that if applicable you deposit your laboratory protocols in protocols.io, where a protocol can be assigned its own identifier (DOI) such that it can be cited independently in the future. For instructions see: http://journals.plos.org/plosone/s/submission-guidelines#loc-laboratory-protocols

We look forward to receiving your revised manuscript.

Kind regards,

Stavros I. Dimitriadis

Academic Editor

PLOS ONE

Additional Editor Comments (if provided):

The reviewer raised issues about the description and also the choice of the methodology proposed by the authors.

I personally agree with the comments and I suggest to revise it according to his/her guidelines.

Journal Requirements:

2. Your ethics statement must appear in the Methods section of your manuscript. If your ethics statement is written in any section besides the Methods, please move it to the Methods section and delete it from any other section. Please also ensure that your ethics statement is included in your manuscript, as the ethics section of your online submission will not be published alongside your manuscript.

Reviewers' comments:

Reviewer's Responses to Questions

**Comments to the Author**

1. Is the manuscript technically sound, and do the data support the conclusions?

Reviewer #1: Yes

2. Has the statistical analysis been performed appropriately and rigorously? 

Reviewer #1: Yes

3. Have the authors made all data underlying the findings in their manuscript fully available?

Reviewer #1: Yes

4. Is the manuscript presented in an intelligible fashion and written in standard English?

Reviewer #1: Yes

5. Review Comments to the Author

Reviewer #1: In this paper, the authors use non-semantic graph analysis to analyze the structure of REM and non-REM dream reports. They extract measures of graph connectedness from these dream reports to determine whether REM and non-REM dream reports differ in terms of their structures, whether these measures outperform the traditional measure of total recall count (TRC), whether these measures and TRC can complement each other in predicting sleep stage, and whether any of the measures relate to a measure of dream complexity. The authors conclude that certain graph measures can complement the use of TRC in predicting sleep stages, and since this graph analysis is simple and automatic, it could be useful in future studies.

While this paper is well motivated and suggests a potentially useful method, we have concerns mostly regarding aspects of the graph analysis and how these results are interpreted. Major and minor comments outlined below.

Major comments:

1. We are concerned about the theoretical interest of LCC/LSC. Inspection of Fig1 suggests that these measures are largely determined by the presence of common conjunctions, prepositions, and pronouns (e.g., “and”, “of”, “I”). Is it true that most of the intersection points are on these words? If so, the results could simply indicate that people use more conjunctions after REM awakenings.

This relates to the possibility mentioned in the discussion (p. 27) that the differential relationships between graph connectedness and PIRs in REM and non-REM dream reports could potentially be explained by increased sleep inertia after non-REM sleep, or differences in how the dream is reconstructed in language due to differences of cognitive state. The authors argue against this because controlling for PIRS eliminates the ability of TRC or LCC to predict sleep stage, but since PIRS data is also collected after awakenings it could similarly be affected by the (in)ability to construct a dream narrative.

2. TRC is framed as contrasting with graph-based measures, but total recall count (TRC) should be highly correlated with the total number of nodes in a full dream graph. Correlations between these measures and between all measures of interest should be reported.

3. The selection of the chosen graph measures should be more strongly justified—for example, in Mota et al. (2014) [ref. 34], graph connectedness was also assessed with average shortest path, average degree, and average clustering coefficient. These measures average across nodes in a graph, therefore adjusting for graph size. Use of these measures might obviate the need to use sliding 30-word windows, which restrict the analysis to the structure of subgraphs and can’t capture the global graph structure; this approach also likely underweights the structures at the beginning and end of the dream report, which could be informative.

4. For the LCC/LSC analysis, if the graphs are derived from temporal word relationships in dream reports, why would the graph be split into different components (e.g., a large component and small component, Fig1A)? Presumably every word is either preceded or followed by another, so this isn’t clear. Is there a pause length that was used to separate the components?

5. We are confused as to the purpose of the “random-likeness” measure. It appears to be a permutation analysis in which the authors generate a null distribution of LCC/LSC measures and compare the actual LCC/LSC measures to this null distribution, which could be used to determine whether the graph connectedness is larger than that expected by chance. Calling this a “random-likeness” measure rather than a permutation test is confusing. Since the authors subtract the mean of the null distribution from the actual values, this implies they are trying to standardize their measures in some way. If this is the case, then why not use these standardized measures throughout the whole experiment, in place of the original LCC/LSC measures?

Minor comments:

1. The authors make clear that they are using non-semantic graphs to analyze the dream reports, but do not explain why semantic graph analysis isn’t used. The highlighted distinction between narrative and conceptual content in REM and non-REM dreams, especially the descriptors “vivid, bizarre, and emotional” in the intro, suggests that interesting semantic differences might be found. Additional justification of the non-semantic approach would be useful.

2. The authors define LCC as “the number of nodes in the maximal component in which all nodes are connected to one another” (p. 11), but this is misleading as it could be interpreted to mean a fully-connected sub-graph where every node is connected to every other node. Mota et al. (2014) define it as a component in which “all pairs of nodes are reachable in an undirected subgraph” and a similar description could be used here. The definition of LSC could similarly be clarified.

3. It would be useful to disambiguate their measure of dream report complexity and network complexity, especially since network complexity likely corresponds with increased network connectedness, and the current results include a negative relationship between network connectedness and dream report complexity.

4. As it is, there are no result figures, and results are displayed in multiple tables with a large number of regression analyses, and it is hard to pull out the meaningful or important results. The descriptive statistics in Table1 could be displayed in a bar graph format.

5. In Figure 1, the font size for the word labels on the graph nodes is too small, especially in B and C.

6. The authors could consider adding a figure to help describe the methods used to create and analyze the random graphs.

6. PLOS authors have the option to publish the peer review history of their article (what does this mean?). If published, this will include your full peer review and any attached files.

Reviewer #1: No

---

## [Author Response · Author response to Decision Letter 0]

3 May 2020

Major Revisions

1. We are concerned about the theoretical interest of LCC/LSC. Inspection of Fig1 suggests that these measures are largely determined by the presence of common conjunctions, prepositions, and pronouns (e.g., “and”, “of”, “I”). Is it true that most of the intersection points are on these words? If so, the results could simply indicate that people use more conjunctions after REM awakenings.

The Referee has a valid concern. In fact, conjunctions, prepositions, and pronouns tend to reduce the size of loops, and may have an indirect effect of LCC/LSC. In the revised manuscript we provide a follow-up analysis where we re-run the graph analyses using a standard list of NLTK stop-words (https://gist.github.com/sebleier/554280). For ease of interpretation for the reader and given this is to rule out a potential confound in our findings, we only re-ran the significant results from our original analysis (this also applies to the analysis pertaining to major revision 4). We were able to reproduce our findings with comparable results: despite a reduced sample size, all findings were either still found to be significant (11/13), or still demonstrated a trend towards significance in the same direction (2/13). Effect sizes were comparable, and in some cases were found to be even stronger than before (e.g. in predicting dream complexity). Given that our core findings were replicated following this control, we interpret this to rule out an increased occurrence of common words as a potential confound for the present results.

The tables reflecting the findings can be found in the supplementary material, while the code can be found in the updated R Notebook. The changes in the manuscript reflect below the main analysis and point the reader to the supplementary material for the corresponding tables.

This relates to the possibility mentioned in the discussion (p. 27) that the differential relationships between graph connectedness and PIRs in REM and non-REM dream reports could potentially be explained by increased sleep inertia after non-REM sleep, or differences in how the dream is reconstructed in language due to differences of cognitive state. The authors argue against this because controlling for PIRS eliminates the ability of TRC or LCC to predict sleep stage, but since PIRS data is also collected after awakenings it could similarly be affected by the (in)ability to construct a dream narrative.

We agree with the Reviewer’s reasoning here that PIRS may also be similarly confounded by sleep inertia and differences in post-awakening cognitive state, and thus it may not be completely ruled out. In the revised manuscript we moderate the argument and describe sleep inertia and dream report complexity as two non-mutually exclusive factors that may explain the present findings, rather than arguing for the latter. We still think it is important to show that, even though sleep inertia cannot be ruled out, the statistical relationship between graph connectedness, sleep stage and PIRS ratings is important to report on. In this regard the partialling out analyses remain useful (e.g. graph connectedness predicts PIRS beyond controlling for differences in sleep stage).

2. TRC is framed as contrasting with graph-based measures, but total recall count (TRC) should be highly correlated with the total number of nodes in a full dream graph. Correlations between these measures and between all measures of interest should be reported.

We thank the Reviewer for the useful recommendation. The revised manuscript includes a table in the supplementary material, including correlations between graph measures (original graph, sliding window and random graph), TRC, as well as simple quantitative report features of interest, such as word count and number of pauses. As the reviewer correctly points out, TRC is highly correlated with attributes in the full dream graph (Spearman’s rho, Edges = .681, LCC = .729 and LSC = .729), which drops substantially once the sliding window method is utilized (Spearman’s rho, Edges = .389, LCC = .257 and LSC = .224). We hope the added table provides sufficient information for this requirement. 

3. The selection of the chosen graph measures should be more strongly justified—for example, in Mota et al. (2014) [ref. 34], graph connectedness was also assessed with average shortest path, average degree, and average clustering coefficient. These measures average across nodes in a graph, therefore adjusting for graph size. Use of these measures might obviate the need to use sliding 30-word windows, which restrict the analysis to the structure of subgraphs and can’t capture the global graph structure; this approach also likely underweights the structures at the beginning and end of the dream report, which could be informative.

We beg to differ. We chose Edges, LCC and LSC because they have been consistently the most informative measures of connectedness in previous literature [1-4]. The approach is thus simple and directed, using only measures that have been of most theoretical interest for the current study. This is also in line with more recent publications that have left out other graph measures and focused on attributes of connectedness and randomlikeness [3] . We agree that other measures may well be interesting and informative, but we feel that their inclusion may further clutter and complicate an already rather complex results section and also add further statistical controls for multiple comparisons, which could lead to an increase in Type I error. We thus wish to maintain and focus on the variables that were originally chosen.

4. For the LCC/LSC analysis, if the graphs are derived from temporal word relationships in dream reports, why would the graph be split into different components (e.g., a large component and small component, Fig1A)? Presumably every word is either preceded or followed by another, so this isn’t clear. Is there a pause length that was used to separate the components?

We thank the Reviewer for the useful recommendation. Pauses here reflect changes in dialogue between the researcher and participant. The revised methods make it clear that separate responses in the dialogue may result in different components - but not necessarily so, since the re-occurrence of a word in separate responses will link the components, as explained in Mota et. al. [5]. Please note also that while this means that pauses will be related to aspects of graph structure, the number of pauses does not directly equate to the number of nodes in LCC, since it is performed across the sliding window and not the entire graph. To rule out the possibility that our findings may merely be reflected by differences in the number of pauses, we performed a supplementary analysis partialling out the number of pauses before evaluating the ability of graph measures to predict differences in sleep stage and dream complexity (see Supplementary Analysis 2). Once again we were able to replicate our core findings: the addition of graph structure predictors was still found to improve the overall fit of the respective models. There were two exceptions to this: firstly, LCC failed to complement TRC in predicting sleep stage, while LSCz failed to predict differences in dream complexity. Nonetheless, similar to our other supplementary analysis, both of these cases demonstrated a significant trend in the same direction (LCC, p = .062; LSCz, p = .061). Thus, overall we feel that this is sufficient evidence against the interpretation that our findings can be explained by a confound deriving from differences in the number of pauses. The manuscript was revised accordingly.

5. We are confused as to the purpose of the “random-likeness” measure. It appears to be a permutation analysis in which the authors generate a null distribution of LCC/LSC measures and compare the actual LCC/LSC measures to this null distribution, which could be used to determine whether the graph connectedness is larger than that expected by chance. Calling this a “random-likeness” measure rather than a permutation test is confusing. Since the authors subtract the mean of the null distribution from the actual values, this implies they are trying to standardize their measures in some way. If this is the case, then why not use these standardized measures throughout the whole experiment, in place of the original LCC/LSC measures?

The purpose of the random-likeness measure is to provide another measure of graph structure through calculating how each graph approximates to a random like structure through comparing it to a distribution of random graphs generated through word shuffling, or permutation. It is not meant to replace graph connectedness, but rather provides an additional complementary method to evaluating graph structure which has been shown to be useful in previous applications (e.g. diagnosis of schizophrenia based on dream reports: [1-4]). Direct evidence for their usefulness as complementary measures can be found in Mota et al. [3], where a linear combination of both connectedness and random-likeness attributes of speech classified negative symptoms and schizophrenia-diagnosis six weeks in advance. Given this empirical grounding, we feel that both of their inclusion is justified in the present study. We have adjusted the methods section to make this justification explicit. 

Minor comments:

1. The authors make clear that they are using non-semantic graphs to analyze the dream reports, but do not explain why semantic graph analysis isn’t used. The highlighted distinction between narrative and conceptual content in REM and non-REM dreams, especially the descriptors “vivid, bizarre, and emotional” in the intro, suggests that interesting semantic differences might be found. Additional justification of the non-semantic approach would be useful.

Perhaps the way we phrased it gave the reviewer the idea that we could either have utilised a semantic or non-semantic graph approach, and that we opted for the latter. The current approach is not related to whether a semantic or non-semantic approach is optimal for investigating dreams per se, but rather is due to the demonstrated usefulness of the current non-semantic approach to “mind-mapping” based on dream reports in cognitive development [1-4]. Semantic approaches to dreaming are indeed very interesting and have been investigated elsewhere [6]. To avoid confusion, we have changed the title from “Non-semantic word graph analysis” to “Word graph analysis” in the methods section, and adjusted the caption in Figure 1 accordingly.

2. The authors define LCC as “the number of nodes in the maximal component in which all nodes are connected to one another” (p. 11), but this is misleading as it could be interpreted to mean a fully-connected sub-graph where every node is connected to every other node. Mota et al. (2014) define it as a component in which “all pairs of nodes are reachable in an undirected subgraph” and a similar description could be used here. The definition of LSC could similarly be clarified.

We thank the Reviewer for the recommendation. The LCC and the LSC will be defined as suggested in Mota et al. [2].

3. It would be useful to disambiguate their measure of dream report complexity and network complexity, especially since network complexity likely corresponds with increased network connectedness, and the current results include a negative relationship between network connectedness and dream report complexity.

Please note that the current results indicate a positive relationship between connectedness and dream report complexity (i.e. larger LCC and LSC are related to more complex mentation). Yet, we agree with the Reviewer that dream report complexity and network complexity are measures that share the label of “complexity” but refer to distinct characteristics of the dream report that may bear no relation to one another. To avoid ambiguity, in the revised manuscript we explicitly state the difference between the two and emphasise how we wish to use the term in the article. (see below):

“The scale was developed as a measure of overall dream quality and quantity, which we infer here to represent the overall complexity of the dream report. It is important to distinguish the term “Dream Report Complexity” as it is used here from the term “Network Complexity”, which can be used to describe the presence and extent of non-trivial characteristics present in a given graph/network. While both of these terms share the label of “complexity”, they describe different aspects of the dream report that may bear no relation to one another. To avoid ambiguities, where the term “Dream Report Complexity” is used, we wish to refer to the complexity of the mentation described by the dreamer as is operationalised by the PIRS.”

4. As it is, there are no result figures, and results are displayed in multiple tables with a large number of regression analyses, and it is hard to pull out the meaningful or important results. The descriptive statistics in Table1 could be displayed in a bar graph format.

We thank the Reviewer for the valuable recommendation, which was duly followed. The tables of the revised manuscript were simplified and the different measures were sorted under sub-headings in the tables to make it easier to identify and pick out the results of interest. Table 1 and 2 was converted into a bar graph figure (Figure 4).

5. In Figure 1, the font size for the word labels on the graph nodes is too small, especially in B and C.

Point taken. The recommendation was implemented via splitting Figure 1 into three separate figures with higher resolution (Fig. 1-3).

6. The authors could consider adding a figure to help describe the methods used to create and analyze the random graphs.

We thank the Reviewer for the recommendation. In the revised manuscript, we edited and elaborated Figure 3 into a flow-chart to make it clearer how the random scores were derived. 

References

1. Mota NB, Vasconcelos NA, Lemos N, Pieretti AC, Kinouchi O, Cecchi GA, Copelli M, Ribeiro S. Speech graphs provide a quantitative measure of thought disorder in psychosis. PloS one. 2012; 7(4).

2. Mota NB, Furtado R, Maia PP, Copelli M, Ribeiro S. Graph analysis of dream reports is especially informative about psychosis. Scientific reports. 2014; 15;4:3691.

3. Mota NB, Copelli M, Ribeiro S. Thought disorder measured as random speech structure classifies negative symptoms and schizophrenia diagnosis 6 months in advance. npj Schizophrenia. 2017; 13;3(1):1-0.

4. Mota NB, Sigman M, Cecchi G, Copelli M, Ribeiro S. The maturation of speech structure in psychosis is resistant to formal education. npj Schizophrenia. 2018; 4(1):1-0.

5. Mota, N. B., Weissheimer, J., Madruga, B., Adamy, N., Bunge, S. A., Copelli, M., & Ribeiro, S. (2016). A naturalistic assessment of the organization of children's memories predicts cognitive functioning and reading ability. Mind, Brain, and Education, 10(3), 184-195.

6. Altszyler E, Ribeiro S, Sigman M, Slezak DF. The interpretation of dream meaning: Resolving ambiguity using Latent Semantic Analysis in a small corpus of text. Consciousness and cognition. 2017; 56:178-87.

---

## [Decision Letter · Decision Letter 1]

1 Jun 2020

PONE-D-20-01935R1

Structural differences between REM and non-REM dream reports assessed by graph analysis

PLOS ONE

Dear Dr. Ribeiro,

Thank you for submitting your manuscript to PLOS ONE. After careful consideration, we feel that it has merit but does not fully meet PLOS ONE’s publication criteria as it currently stands. Therefore, we invite you to submit a revised version of the manuscript that addresses the points raised during the review process.

Reviewers raised a few more comments regarding the formatting of the draft, the consistency between main draft and supplementary material and further correction of the figures's captions.

I encouraged you to address them one by one and re-submit the revised manuscript.

We look forward to receiving your revised manuscript.

Kind regards,

Stavros I. Dimitriadis

Academic Editor

PLOS ONE

Additional Editor Comments (if provided):

After carefully read your draft and the comments from the reviewers, I encouraged you to address them and re-submit

a new revised manuscript.

Their comments focuses on:

1) shortening of specific parts of the draft

2) improve the structure of the draft with appropriate numbering

3) align the report within the draft with the information in supplementary material.

Reviewers' comments:

Reviewer's Responses to Questions

**Comments to the Author**

1. If the authors have adequately addressed your comments raised in a previous round of review and you feel that this manuscript is now acceptable for publication, you may indicate that here to bypass the “Comments to the Author” section, enter your conflict of interest statement in the “Confidential to Editor” section, and submit your "Accept" recommendation.

Reviewer #1: (No Response)

Reviewer #2: All comments have been addressed

2. Is the manuscript technically sound, and do the data support the conclusions?

Reviewer #1: Yes

Reviewer #2: Yes

3. Has the statistical analysis been performed appropriately and rigorously? 

Reviewer #1: Yes

Reviewer #2: Yes

4. Have the authors made all data underlying the findings in their manuscript fully available?

Reviewer #1: Yes

Reviewer #2: Yes

5. Is the manuscript presented in an intelligible fashion and written in standard English?

Reviewer #1: Yes

Reviewer #2: Yes

6. Review Comments to the Author

Reviewer #1: The authors have addressed the main concerns thoroughly. Appropriate control analyses have been added, and methods and measures have been clarified. Some remaining minor concerns listed below:

- It appears that “number of paragraphs” and “number of pauses” reflect the same measure, but that has not been made clear in the manuscript. For example, the follow-up analyses control for number of paragraphs, but Supplementary Table 2 includes statistics for number of pauses. The authors should make clear whether these are in fact the same measure, or if they reflect different aspects of the graphs.

- The authors ran a follow-up analysis to partial out number of paragraphs before assessing graph-based differences. To further motivate this analysis, it would be useful to include descriptive and comparison stats on number of paragraphs observed in the different sleep stages.

- Figure 2: Caption says window size is 15 words and overlap is 5 words, but the figure seems to show an overlap of 10 words.

- Figure 3: Why are some words red in the shuffled dream reports? There does not seem to be a need to highlight shuffled words, and this is confusing since nodes of graph are also red. Also, the abbreviations used in the figure (i.e., mrLCC, sdrLCC, mrLSC, sdrLSC) should be explained in the caption.

Reviewer #2: Many thanks to the authors for their detailed response and revision of the initial manuscript. All the valid points previously raised were addressed and an appropriate adjustment or rebuttal was provided. As such, I do not have any additional specific criticisms to mention. The revised version of the manuscript reads well and would be a meaningful addition to the literature of how to analyze dream reports. As an overarching impression, it could be shortened, particularly aspects such as the intro that expand to 5 whole pages and appear better suited for a master thesis than a journal paper. Each section and subsection can be numbered to facilitate reading. Finally, as this was a proof of concept study in healthy controls, it would be nice to incorporate some clinical repercussions in the end of the discussion (for example using such methods to evaluate patients with REM behavior or other sleep disorders and/or medication effects to dreams).

7. PLOS authors have the option to publish the peer review history of their article (what does this mean?). If published, this will include your full peer review and any attached files.

Reviewer #1: No

Reviewer #2: No

---

## [Author Response · Author response to Decision Letter 1]

22 Jun 2020

Reviewer 1 Comments

- It appears that “number of paragraphs” and “number of pauses” reflect the same measure, but that has not been made clear in the manuscript. For example, the follow-up analyses control for number of paragraphs, but Supplementary Table 2 includes statistics for number of pauses. The authors should make clear whether these are in fact the same measure, or if they reflect different aspects of the graphs.

We thank the reviewer for pointing this out. Pauses and number of paragraphs here do refer to the same measure, although we decided to opt for the latter since it reflects turn-taking between the researcher and the participant (i.e. dreamer), as opposed to pauses in their speech per se. We have now changed the item in the supplementary table 2 to reflect “No. Paragraphs”, to make it clear that they refer to the same measure as before. 

- The authors ran a follow-up analysis to partial out number of paragraphs before assessing graph-based differences. To further motivate this analysis, it would be useful to include descriptive and comparison stats on number of paragraphs observed in the different sleep stages.

We thank the reviewer for the suggestion. We now provide basic comparisons for number of paragraphs in the results section to further justify the need for this analysis. 

- Figure 2: Caption says window size is 15 words and overlap is 5 words, but the figure seems to show an overlap of 10 words.

The reviewer is correct, thanks for pointing out this mistake. The figure does indeed show an overlap of 10 words, not 5. We have corrected this accordingly.

- Figure 3: Why are some words red in the shuffled dream reports? There does not seem to be a need to highlight shuffled words, and this is confusing since nodes of graph are also red. Also, the abbreviations used in the figure (i.e., mrLCC, sdrLCC, mrLSC, sdrLSC) should be explained in the caption.

We originally thought it may be useful to show which of the words had been shuffled in the resulting dream report. However, we agree with the reviewer that this is not necessary and may also be confusing due to the colouring of nodes. We have changed the highlighted text to black in figure 3 and have provided attached explanations for the abbreviations as suggested. 

Reviewer 2 Comments

Reviewer #2: Many thanks to the authors for their detailed response and revision of the initial manuscript. All the valid points previously raised were addressed and an appropriate adjustment or rebuttal was provided. As such, I do not have any additional specific criticisms to mention. The revised version of the manuscript reads well and would be a meaningful addition to the literature of how to analyze dream reports. As an overarching impression, it could be shortened, particularly aspects such as the intro that expand to 5 whole pages and appear better suited for a master thesis than a journal paper. Each section and subsection can be numbered to facilitate reading. Finally, as this was a proof of concept study in healthy controls, it would be nice to incorporate some clinical repercussions in the end of the discussion (for example using such methods to evaluate patients with REM behavior or other sleep disorders and/or medication effects to dreams).

We thank the reviewer for the helpful feedback and additional suggestions. Where possible we have tried to lower the word count to make it more readable, specifically in the introduction section. We have also added a brief section at the end to reflect possible applications to psychopathology. Perhaps this is just a point of preference, but it does not seem that in general PLoS ONE articles use numbering for the sub-sections. As a result, we prefer to leave them without numbers to be standardised in terms of formatting with the other articles.

---

## [Editor Report · Decision Letter 2]

26 Jun 2020

Structural differences between REM and non-REM dream reports assessed by graph analysis

PONE-D-20-01935R2

Dear Dr. Ribeiro,

We’re pleased to inform you that your manuscript has been judged scientifically suitable for publication and will be formally accepted for publication once it meets all outstanding technical requirements.

Kind regards,

Stavros I. Dimitriadis

Academic Editor

PLOS ONE

Additional Editor Comments (optional):

Reading carefully the comments raised by the reviewers and your answer, I agree that your draft

has passed the quality control via the reviewing process.

The manuscript is more readable than the original version.

I recommend the acceptance of the manuscript.
---

## [Editor Report · Acceptance letter]

6 Jul 2020

PONE-D-20-01935R2 

Structural differences between REM and non-REM dream reports assessed by graph analysis 

Dear Dr. Ribeiro:

I'm pleased to inform you that your manuscript has been deemed suitable for publication in PLOS ONE. Congratulations! Your manuscript is now with our production department. 

Kind regards, 

on behalf of

Dr. Stavros I. Dimitriadis 

Academic Editor

PLOS ONE